# Impact of Using Oilseed Industry Byproducts Rich in Linoleic and Alpha-Linolenic Acid in Ruminant Nutrition on Milk Production and Milk Fatty Acid Profile

**DOI:** 10.3390/ani14040539

**Published:** 2024-02-06

**Authors:** Bojana Kokić, Slađana Rakita, Jelena Vujetić

**Affiliations:** Institute of Food Technology, University of Novi Sad, Bulevar cara Lazara 1, 21000 Novi Sad, Serbia; sladjana.rakita@fins.uns.ac.rs (S.R.); jelena.vujetic@fins.uns.ac.rs (J.V.)

**Keywords:** hempseed byproducts, pumpkin seed byproducts, sunflower seed byproducts, camelina seed byproducts, linseed byproducts, dairy ruminants, rumenic acid, vaccenic acid

## Abstract

**Simple Summary:**

The fatty acid composition of milk fat plays a crucial role in determining the nutritional quality of milk because particular fatty acids can have positive or negative influences on human health. Production of milk with a lower content of saturated fatty acids, and a higher content of unsaturated fatty acids, particularly n-3 fatty acids, is preferable due to their beneficial effects on health. Additionally, a higher level of rumenic and vaccenic acid in milk is desirable, as these fatty acids are believed to impart health-related properties. Manipulating animal nutrition proves most effective in achieving the desired milk fatty acid composition, with the inclusion of unconventional feed ingredients like oilseed processing byproducts being a successful strategy for modifying milk fatty acid profiles. This review provides a comprehensive overview of the existing 41 research studies regarding the incorporation of hemp, pumpkin, sunflower, camelina, and linseed byproducts into the nutrition of dairy cows, sheep, and goats, and their influence on milk production traits and milk fatty acid composition. This review concludes that, overall, the selected oilseed byproducts exhibit promising potential to improve the fatty acid profile of milk, with no adverse effects on milk production and composition.

**Abstract:**

Milk contains more than 400 different fatty acids, some of which play a positive role in promoting human health. The profile of fatty acids in milk can be enhanced by providing animals with plant-based resources that possess feeding characteristics adequate for favorable changes in the fatty acid composition and increasing healthy fatty acids in milk. This review summarizes the available 41 research studies on the utilization of oilseed industry byproducts rich in linoleic acid (hemp, pumpkin, sunflower) and alpha-linolenic acid (camelina and linseed) in dairy cow, sheep, and goat nutrition; their impact on milk production characteristics; and potential to improve fatty acid composition of milk through the diet. This review illustrates that incorporating byproducts into the diet for dairy ruminants generally does not have any adverse effects on both milk production and composition. A similar trend of improvement in milk fatty acid profile was observed when ruminants were fed diets supplemented with camelina, linseed, and sunflower byproducts, while no significant changes were noted with pumpkin byproducts. Hempseed byproducts showed potential for use as an alternative ingredient in dairy ruminant diets. Nevertheless, more in-depth research investigating the inclusion of selected byproducts is required before valid conclusions can be drawn regarding their value.

## 1. Introduction

Consumer awareness of the connection between health and diet is on the rise, making the nutritional quality of food a widely discussed subject. This has led to a growing interest in functional foods, which are foods or food components that offer health benefits beyond their fundamental nutritional value, and they are gaining popularity in the market.

Milk serves as a significant source of dietary energy in human nutrition, containing numerous essential nutrients such as protein, lipids, lactose, vitamins, and minerals. Nutritional value, physical, and sensory characteristics of milk originate from fats. Milk fat is recognized as one of the most complex naturally occurring fats, with an estimated composition of more than 400 different fatty acids (FAs) [1]. Milk fat is a crucial component that significantly impacts the processing of raw materials and plays a pivotal role in carrying taste and aroma. The fat content in cow’s milk typically falls within the range of 3.3% to 4.4%, while goat’s milk and sheep’s milk contain approximately 3.2% to 4.2% and 7.1% fat, respectively [2].

The processes of dietary lipid metabolism in ruminants, as well as the biosynthesis of milk fat, are highly complex. In brief, when dietary lipids enter the rumen, the initial step involves extensive hydrolysis carried out by rumen bacteria, leading to the release of free FAs [3]. The next significant transformation of dietary fats in the rumen is the biohydrogenation of unsaturated fatty acids (UFAs) into saturated fatty acids (SFAs). Only a select few species of rumen bacteria participate in this process, which serves as a protective mechanism against the toxic effects of UFAs [4]. In ruminant milk, SFAs make up approximately 60–70% of the total, monounsaturated fatty acids (MUFAs) constitute around 20–35%, and polyunsaturated fatty acids (PUFAs) represent as little as 3% of the overall FA composition [2]. Milk FAs originate from two primary sources, *de novo* synthesis in the mammary gland and the uptake of pre-existing long-chain FAs by the mammary tissue [5]. Although the diet of ruminants contains a significant proportion of UFAs, the extensive breakdown and transformation of these UFAs in the rumen result in the majority of FAs reaching the small intestine being predominantly free SFAs. Nonetheless, certain biohydrogenation intermediates, including rumenic acid (RA) and vaccenic acid (VA), can escape from the rumen. RA is produced as an intermediate solely in the biohydrogenation of linoleic acid (LA), whereas VA is an intermediate in the biohydrogenation of both LA and alpha-linolenic acid (ALA) [6].

Conjugated linoleic acid (CLA) is a group of naturally occurring isomers of LA that differs in the position or geometry (i.e., cis or trans) of their double bonds. The major source of CLA in the human diet is through the consumption of red meat and dairy products, while smaller quantities can be found in poultry, seafood, and vegetable oil [7]. In recent decades, CLA has garnered significant attention due to its health-related properties, which include its potential as an anticarcinogenic, anti-diabetic, anti-inflammatory, anti-obesity, and anti-atherogenic compound [8,9,10]. Milk CLA is derived from CLA produced in the rumen, but a major source of it comes from the endogenous synthesis of CLA in the mammary gland via a pathway related to the desaturation of VA by the Δ9-desaturase [11]. Rumenic acid (RA, cis-9,trans-11 CLA) is the major isomer of CLA in ruminant-derived food products and represents greater than 80% of the CLA present in milk fat [12]. Vaccenic acid (VA, trans-11 C18:1) is a positional and geometric isomer of oleic acid, and the major source of this trans FA in human diet is ruminant fat. Dietary VA can be desaturated to RA in ruminants, rodents, and humans. Bauman and Lock [6] emphasized that the functional food aspects of CLA isomers in dairy products primarily focus on RA as the major isomer, while also underscoring the importance of including VA in this consideration due to its role in the endogenous synthesis of RA in the human body.

Consequently, numerous studies have focused on enhancing the value of animal products, particularly by increasing the concentration of FAs in milk and dairy products that have favorable effects on human health [13,14,15,16,17]. From a nutritional perspective, a more favorable FA profile would entail a reduced proportion of SFAs and an increased proportion of UFAs, with particular focus on n-3 PUFAs, along with a lower n-6/n-3 ratio [18]. Modifying the FA composition of milk provides advantages for the human diet without requiring modifications in consumer dietary preferences. This approach preserves the nutritional advantages linked to the macronutrients and micronutrients present in dairy products.

Factors influencing the FA composition of milk can be categorized as biological (such as breed, individual cow variations, milk yield, lactation stage, and parity) and external (including the composition of the feed and management practices). While all these factors can impact the FA profile of milk, nutrition-related factors are of primary significance. In most ruminant diets, fat constitutes less than 5% of the total dry matter (DM) [19]. Initially, the addition of fat to diets was primarily aimed at enhancing the energy content to meet the energy needs of dairy ruminants. Yet, it has been discovered that fat supplementation serves additional functions, impacting the FA profiles of both meat and milk in ruminants. However, the supplementation of fat in ruminant diets, especially with C18 PUFA, is restricted due to its adverse effects on ruminal bacterial growth [20]. As highlighted by Fredeen [21], lipid supplements rich in PUFAs can have an antimicrobial effect, resulting in reduced fiber digestion, a decreased acetate-to-propionate ratio, and a decrease in milk fat synthesis.

Considering that RA is exclusively produced as an intermediate in the biohydrogenation of LA, while VA is an intermediate in the biohydrogenation of both LA and ALA, we chose five oilseed industry byproducts (BPs) rich in LA (hemp, pumpkin, sunflower) and ALA (camelina and linseed). This review provides a summary of studies that explore the inclusion of selected oilseed industry BPs in dairy ruminant nutrition, their impact on milk production traits, and potential to enhance the FA composition of ruminant milk through diet.

## 2. Oilseed Plants: General Characteristics and Their Byproducts

Hemp (*Cannabis sativa*) is considered an environmentally friendly crop due to its high productivity, and resistance to drought, pests, and disease. It can be grown with minimal impact on the ecosystem, it prevents soil erosion, and improves soil quality. Due to a high nutritional value and functional features, hemp can be used in a wide range of applications, including textiles, paper, biodegradable plastics, medical and pharmaceutical products, food products, animal feed, and more [22].

Pumpkin (*Cucurbita pepo*) has been recognized as a pioneering crop, providing a balanced diet, and displaying better adaptation to less favorable soil and atmospheric conditions in comparison to other crops. The different species of pumpkin are grown for their seeds and fruits, serving primarily as a nutritious food source that provides high-quality oil and valuable nutrients like protein, essential fatty acids, and dietary fiber [23].

Sunflower (*Helianthus annuus*) is one of the important oilseed crops grown for edible oil production and biodiesel production [24]. The adaptability of sunflower to diverse environmental conditions has significantly expanded its cultivation as a prominent oilseed crop throughout the world. Due to its beneficial effects on human health, sunflower has been used as a functional food or nutraceutical [25].

Camelina (*Camelina sativa*) is an ancient plant which has been cultivated in the past; however, it has gained increasing popularity in the last decade thanks to its distinctive nutritional profile and agronomic attributes, which include adaptability to diverse environmental conditions, low fertilizer requirements, relatively short growing season, and good resistance to diseases and pests [26,27]. It has been considered a sustainable and renewable raw material for biofuel production due to its abundant oil content [28] and promising alternative ingredient in animal nutrition [29].

Throughout history, linseed (*Linum usitatissimum L.*) has been identified as an invaluable plant for cultivation, primarily owing to its wide array of versatile uses. It has been primarily used for the production of seeds and fibers [30]. Linseed has nutritive properties significant for human nutrition, including high amounts of n-3 essential FAs which demonstrate numerous health benefits [31].

During the extraction process of oil from oilseeds, different BPs such as meal, cake, and expeller are produced. If the oil extraction is carried out using a solvent, the obtained BP is meal, whereas cold-pressed cake is generated during mechanical cold-pressing of oilseed. If oilseed preconditioning is applied before pressing, the resulting BP is called expeller. Inadequate usage of BP terminology is observed in scientific literature; however, as already emphasized by Rakita, et al. [32], the correct use of these terms is important to avoid misinterpretation and enable more reliable comparison of results between different studies. Mechanical and solvent extraction methods contribute to the reduction of fat while concurrently enhancing the content of protein and amino acids in the BPs. Due to their substantial nutritional profile, the oilseed BPs are sought as valuable sources of protein and energy in ruminant nutrition [32]. In 2022, 145.8 million tons of compound feed was produced in the EU, of which 37.2 million tons of cakes and meals, and 18.0 million tons of co-products derived from the food and bioethanol sectors were used as ingredients in the feed industry [33].

## 3. Chemical Composition of Oilseed Byproducts

Table 1 provides an overview of the chemical composition of the selected BPs. As seen, all BPs have similar levels of DM and ash. The oilseed BPs have proven to be an excellent protein source. All observed BPs have a high protein content, particularly BPs from pumpkin seed processing (46.2% mean value). As shown, the BPs exhibit considerable diversity in their fat content (e.g., 1.1–16.3% for sunflower BPs), because fat content varies depending on the process of BP generation. Specifically, oilseed meals are derived by solvent extraction and thus have a lower fat content compared to cakes and expellers obtained via mechanical pressing which retain a higher amount of fat and thus contribute a considerable amount of energy to animal diets. Fiber content significantly varies among the BP, with sunflower seed, hempseed, and pumpkin seed BPs having the higher mean of cellulose content (30.6, 26.1, and 22.2%, respectively) in comparison to camelina and linseed BPs (11.1 and 9.6%, respectively). Sunflower seed and hempseed BPs also show the highest mean NDF and ADF content.

With regards to the FA profile, selected BPs demonstrate a lower share of SFA, ranging between 5.3 and 18.2%, with palmitic acid being the predominant, followed by stearic acid. The highest mean levels of MUFAs are in sunflower and pumpkin seed BPs (39.1 and 32.7%, respectively). Oleic acid is the major MUFA distributed in all BPs. All selected BPs exhibit a substantial level of PUFAs, particularly those of linseed and hempseed (83.9 and 72.8%, respectively). LA (PUFA n-6) and ALA (PUFA n-3) are regarded as essential FA. These essential FAs are crucial for maintaining the health and well-being of animals, as they play roles in various biological processes, including energy metabolism, cell membrane structure, and the regulation of inflammatory responses [34]. Animals must obtain these FAs through their diet since their bodies are not able to synthesize them. Hempseed, pumpkin seed, and sunflower seed BPs are the major source of LA. Regarding ALA, linseed BPs have the highest mean value of this FA (55.8%), followed by camelina seed BPs (36.1%), while hempseed BPs have moderate share of ALA (16.8%). The lowest ratio of n-6 and n-3 FAs are the BPs of camelina seed and linseed (0.7 and 0.4, respectively), followed by hempseed BPs (3.4).

**Table 1 animals-14-00539-t001:** Chemical composition of selected oilseed byproducts.

	**Hemp BP**		**Pumpkin BP**		**Sunflower BP**		**Camelina BP**		**Linseed BP**	
	**Mean**	**Min–Max**	**Mean**	**Min–Max**	**Mean**	**Min–Max**	**Mean**	**Min–Max**	**Mean**	**Min–Max**
Chemical Composition										
Dry matter (% as fed)	91.4	89.4–93.7	89.6	85.8–93.9	92.6	90.0–97.2	90.6	85.8–92.6	91.3	88.3–95.0
Protein (% DM)	33.0	31.1–34.4	46.2	34.9–55.3	30.2	24.3–37.5	36.5	30.5–43.5	34.9	27.8–39.4
Fat (% DM)	11.0	8.9–12.4	13.0	8.2–16.3	7.0	1.1–16.3	13.7	12.9–14.8	10.7	1.4–16.9
Fiber (% DM)	26.1	26.1	22.2	3.9–34.4	30.6	26.0–32.4	11.1	10.2–12.9	9.6	7.6–12.8
NDF (% DM)	41.8	39.3–43.6	29.6	21.4–45.0	42.2	33.9–48.1	24.5	22.0–26.9	26.2	21.3–36.1
ADF (% DM)	33.8	32.1–36.2	15.9	8.1–31.0	28.8	21.2–32.3	14.8	14.4–15.1	15.1	12.1–17.7
Ash (% DM)	6.5	6.3–6.7	6.8	5.3–8.5	5.9	5.2–7.1	5.3	5.3	6.8	4.7–10.7
Main fatty acids (% on total fatty acids)										
Palmitic	8.8	8.2–9.3	10.5	8.2–12.3	6.2	3.9–8.0	8.0	7.2–10.3	5.8	4.0–7.7
Stearic	3.4	2.9–3.8	3.7	1.8–5.2	3.3	1.5–5.0	2.6	2.6	2.2	1.3–3.1
Oleic	12.7	12.2–13.1	32.6	28.8–37.3	39.0	28.0–56.7	16.5	13.1–19.9	17.2	15.9–18.5
Linoleic	54.3	52.5–56.0	51.3	50.6–52.1	50.4	37.7–56.9	25.9	22.3–28.5	18.4	10.2–26.6
Alpha-linolenic	16.8	14.4–19.1	0.3	0.1–0.5	0.8	0.03–2.1	36.1	29.5–41.3	55.8	42.9–68.6
SFA	12.4	11.8–13.1	14.6	10.3–18.2	9.6	5.4–13.1	9.5	7.3–11.7	8.2	5.3–11.1
MUFA	12.8	12.4–13.1	32.7	28.9–37.3	39.1	28.1–56.8	28.4	27.7–29.1	17.3	15.9–18.7
PUFA	72.8	71.8–73.8	63.1	52.4–84.1	51.1	37.7–59.0	61.6	58.2–65.0	83.9	78.8–88.9
n-6	55.8	54.7–56.9	51.9	51.7–52.2	50.4	37.7–56.9	25.4	24.5–26.2	18.8	10.2–27.3
n-3	16.8	14.4–19.1	0.6	0.1–1.1	0.8	0.03–2.1	36.2	33.7–38.8	55.8	42.9–68.6
n-6/n-3	3.4	2.9–4.0	173.9	45.8–366.2	482.5	27.8–1137.4	0.7	0.7	0.4	0.2–0.6

DM—dry matter; NDF—neutral detergent fiber; ADF—acid detergent fiber; SFA—saturated fatty acids; MUFA—monounsaturated fatty acid; PUFA—polyunsaturated fatty acids. Presented values were collected using data: hempseed byproduct [35,36,37]; pumpkin seed byproduct [38,39,40,41,42,43]; sunflower seed byproduct [24,43,44,45,46,47,48,49,50,51]; camelina seed byproduct [28,52,53,54,55,56]; linseed byproduct [42,43,47,57,58,59,60,61,62,63,64].

## 4. Literature Search Strategy

A comprehensive literature search of Google Scholar was performed in January 2023. This process was conducted over a period of three months by the lead author. The literature search was limited to articles published in English until the year 2023, including two papers published in 2023 that were additionally included in the analysis. The search keywords used to collate the literature were: (hempseed OR pumpkin seed OR sunflower OR camelina OR linseed OR flax) AND (meal OR cake OR expeller) AND (milk fatty acid profile OR dry matter intake OR milk yield OR milk protein OR milk fat OR milk lactose). Full papers presenting original data were used for this review, while publications in the form of conference abstracts, conference proceedings, short communications, and systematic reviews were not taken into consideration. In total, 41 relevant articles were included in this scoping review. The collected articles were methodically organized and classified based on the three main factors: (i) ruminant species, (ii), oilseed species, and (iii) oilseed BP form (meal, cake, or expeller).

Figure 1 illustrates the overview of the chronological progression of publications over the years. Within our dataset, it is noteworthy that the earliest paper to openly investigate the use of oilseed BPs in lactating cows first emerged in 1977. Publications addressing the use of sunflower and linseed BPs were distributed evenly throughout the observed timeframe. Publications regarding camelina BPs emerged in 2007, and this research continued to be actively pursued up to today. Studies including hemp and pumpkin BPs were first published in 2010 and 2017, respectively, and following that, the research is sporadically represented.

The majority of studies focus on utilizing oilseed BPs in the diets of dairy cows, with fewer studies addressing their utilization in goat and sheep diets (Figure 2). Studies concerning dairy cows primarily focused on sunflower and linseed BPs, while research on dairy sheep predominantly centered on sunflower BPs. Notably, pumpkin seed BPs were most frequently examined in the diet of dairy goats. However, no research was found regarding the incorporation of pumpkin seed and linseed BPs into the nutrition of dairy sheep.

## 5. Effect of Hemp, Pumpkin, and Sunflower Seed Byproducts on Milk Yield, Composition, and Fatty Acids

A review of the collected literature revealed discrepancies in the chemical composition in the experimental diet compared to the control diet, particularly in the content of protein, fat, fiber, and non-fibrous carbohydrates. To address this, detailed data regarding the dietary ingredients and chemical composition of the control and experimental diets are presented in Appendix A. This table aims to provide a clearer understanding of the variations and inconsistencies in the chemical composition of the diets identified in the collected studies.

### 5.1. Milk Yield and Composition

After reviewing the available literature, only one study was found for each ruminant species regarding the usage of hempseed cake (HSCAKE) (Table 2). In the Karlsson, et al. [35] study, the effects of increasing the proportion of HSCAKE in the dairy cow diet from 0 to 318 g/kg DM were assessed. The maximum production was observed in cows fed 143 g HSCAKE/kg DM, which corresponded to a dietary crude protein concentration of 157 g/kg DM. Beyond this point, when higher amounts of HSCAKE were included in the diet, production decreased, with both milk protein and milk fat concentrations decreasing linearly. The authors also observed that a high inclusion level of HSCAKE, which resulted in a higher fat and NDF content in the experimental ration compared to the control, did not limit feed intake [35]. In the Šalavardić, et al. [37] study, although no statistically significant differences were observed, numerical trends suggested that when goats were fed HSCAKE, milk yield tended to be higher, while milk protein and fat content tended to be lower. In the same study, a lower lactose concentration was reported, aligning with the results from the Mierliță [36] study, which also indicated decreased lactose concentration in the milk from HSCAKE-fed sheep compared to the control group. It is important to note that only one published study, conducted by Mierliță [36], reported statistically significant increases in milk yield (+5.4%) and milk fat content (+7.4%) in dairy sheep when HSCAKE was included in their diet. The authors suggested that the observed increase in milk yield associated with HSCAKE consumption could be attributed to the well-balanced amino acid composition of hemp, which supports milk protein synthesis. Additionally, the significant increase in milk fat content was likely due to the two-fold greater fat content in the experimental diet containing HSCAKE. Considering the limited number of published studies on observed hempseed BPs, further research is needed to confirm the hypothesis regarding the impact of hempseed BPs on milk production parameters.

In recent years, there has been an increasing interest in investigating the possibility of incorporating pumpkin seed cake (PSCAKE) into the diets of dairy ruminants. Several recent studies have examined the potential of using PSCAKE as a replacement for soybean meal [39,40,41,65] and sunflower meal [38] in the diets of dairy cows and goats (Table 2). In all these studies, regardless of the inclusion levels, no detrimental effects were observed on milk production or quality. This indicates that PSCAKE has the potential to be used as a feed source for dairy cows and goats; however, more research should be undertaken before valid conclusions about the potential impact of using PSCAKE in ruminant nutrition can be made.

Given that the industrial use of sunflower seeds for oil production started over a century ago, there has been a notable increase in research efforts spanning more than 40 years that are dedicated to exploring the potential utilization of sunflower BPs in ruminant nutrition. The findings from the initial published study by Schingoethe, et al. [50] demonstrated that there were no significant alterations in either milk yield or milk fat and protein content when soybean meal was substituted for sunflower meal (SFMEAL) in the diet of dairy cows. The same findings were corroborated by others [45,49,66,67], all confirming that the partial or complete replacement of soybean meal with sunflower BPs did not adversely affect cow’s milk production traits. In contrast, Yildiz, et al. [68] reported a significant decrease in milk yield (−9.3%), while Pirlo, et al. [46] observed a decrease in milk protein content (−2.8%). The latter effect could be a result of a dilution effect, as lipid supplementation often increases milk production without changing protein yield. Additionally, in the Oliveira, et al. [24] study, an increase in the proportion of SFMEAL led to a linear decrease in milk yield.

Studies examining the use of sunflower BPs in small ruminant diets have yielded mixed results. In the Antunović, et al. [69] study, sheep fed sunflower cake (SFCAKE) exhibited higher milk fat content (+20.0%) compared to those fed soybean meal, likely due to the greater fat content in the experimental diet containing SFCAKE (34 vs. 29 g/kg DM). Conversely, Amores, et al. [44] observed a significant decrease in milk fat content, which was attributed to the milk fat depression syndrome (induced by the high amount of LA). Other studies [48,70,71] did not find notable changes in either milk yield or quality when sunflower BPs were included in the diets of dairy sheep and goats.

In studies that presented the compositions of treatment diets, an increased fat content was observed in diets supplemented with BPs compared to control diets without BPs [35,36,40,41,44,45,46,48,66,69], while lower fat content was also noted in some studies [24,37,72]. As stated, variations in the dietary and chemical composition between the control and experimental diets were commonly noted in the mentioned studies investigating the impact of dietary inclusion of oilseed BPs on milk composition. Hence, the variations in observed milk production traits might be ascribed to differences in the chemical composition of the diets rather than a direct impact of the BPs themselves [73].

### 5.2. Milk Fatty Acids

Upon reviewing the available literature, only two studies were found that examined the influence of HSCAKE supplementation on the FA profile of milk in dairy sheep [36] and goats [37] (Table 3). In the Mierliță [36] study, HSCAKE was included in the experimental ration as a partial replacement for corn, barley, and SFMEAL. The inclusion of HSCAKE led to several changes in the FA composition of the diet compared to the control, including an increase in stearic acid and ALA, and a decrease in LA. HSCAKE, being a rich source of PUFAs that are toxic to ruminal microorganisms, resulted in a decrease in milk SFA. Simultaneously, MUFA, PUFA, and n-3 FA content in milk increased, along with a significant increase in RA and VA, approximately three-fold and two-fold, respectively. All observed changes in the milk FA profile were desirable from a nutritional perspective. In the Šalavardić, et al. [37] study, the partial replacement of soybean meal and extruded soybean with HSCAKE resulted in several changes in the FA composition of the experimental diet compared to the control. Specifically, there was a reduction in stearic acid, which is known to inhibit *de novo* FA synthesis, mainly C10:0–C16:0. Additionally, there was an increase in ALA. The HSCAKE-supplemented diet led to lower proportions of C18:0 in goat milk compared to the control. This decrease may be attributed to a reduction in the complete biohydrogenation of dietary PUFA to C18:0 and a lower proportion of C18:0 in the HSCAKE feed mixtures. On the other hand, there was a slight increase in milk SFA, primarily due to an increase in saturated FA like C10:0, C11:0, and C14:0. Furthermore, the HSCAKE groups showed a tendency to have a higher proportion of n-3 FA in milk compared to the control group. However, the results indicated that the transfer of n-3 FA from the feed to milk fat was relatively low. This could be explained by the fact that ALA, being one of the FAs most intensively biohydrogenated, had a low apparent transfer efficiency into milk fat. Both of the above-mentioned studies reported a desirable decrease in the milk n-6/n-3 ratio.

A few recent studies were conducted to explore the possibility of replacing soybean meal with PSCAKE in the nutrition of dairy cows [41] and dairy goats [39,65]. Based on the obtained results, the authors concluded that PSCAKE had little effect on the FA content of milk, including RA and VA. This was presumably because most of the oleic acid and LA in the PSCAKE were biohydrogenated by microorganisms after entering the rumen. Additionally, as soybean meal and PSCAKE have similar LA content, with soybean meal containing a slightly higher ALA content, the substitution of soybean meal with PSCAKE did not result in any increase in the 18-carbon PUFA precursors in the diet. On the other hand, when SFMEAL was replaced with PSCAKE, a more favorable milk FA profile was achieved, characterized by lower SFA and higher MUFA and PUFA content, as reported by Boldea, et al. [38]. However, this replacement also led to a significant increase in milk n-6 FA, mainly LA, resulting in an increased n-6/n-3 ratio. In the same study, there was only a tendency to increase milk RA.

Analyzing the data from Table 3, it becomes apparent that introducing SFCAKE into the diet of dairy ruminants leads to more consistent outcomes in terms of milk FA profile compared to HSCAKE and PSCAKE. Both cows [46,67] and sheep [44,69,70] that were fed SFCAKE produced milk with lower SFA and higher MUFA and PUFA content, with the magnitude of this change being higher with increased intake of oil from the BPs. However, in the Pascual, et al. [70] study, SFCAKE increased the milk’s n-6 proportion (primarily LA) to a greater extent, resulting in a significant concurrent increase in the milk’s n-6/n-3 ratio, irrespective of the forage type. The enrichment of milk with RA (up to +155%) and VA (up to +182%) was observed in all of the studies. Goiri, et al. [67] suggested that LA from SFCAKE may have inhibited the last step of biohydrogenation and increased the ruminal outflow of VA.

**Table 2 animals-14-00539-t002:** The impact of incorporating hemp, pumpkin, and sunflower byproducts into the diets of dairy ruminants on dry matter intake, milk yield, protein percentage, fat percentage, and lactose percentage. The data are presented as the proportionate difference between the treatment group, considering the specific level of inclusion, and the control group.

Ruminant Species	Byproduct Form ^1^	Inclusion in Diet (g/kg DM Unless Otherwise Stated)	Intake of BP (kg DM/Day Unless Otherwise Stated)	Intake of Fat (g/Day) ^2^	Number of Animals	Trial Duration	Main Feed Ingredient Replaced ^3^	F:C ^4^	DMI ^5^	Milk Yield	Milk Protein	Milk Fat	Milk Lactose	Reference
									% Change—Relative to Control ^6^					
Cow	HSCAKE	143	3.78 *	468 *	40	5 weeks	compound pellets	50:50	13.3	13.9	−0.6	−2.3	0.9	[35]
		233	5.57 *	690 *					2.6	6.3	−3.9	−5.6	2.6	
		318	8.40 *	1041 *					13.3	6.3	−6.3	−9.7	−5.8	
Sheep	HSCAKE	213 as fed *	0.48 as fed	50 *	30	10 weeks	SFM; C; B	60:40	1.6	**5.4**	*0.2*	**7.4**	**−6.7**	[36]
Goat	HSCAKE	60 (in conc) ^7^	0.07 *	6 *	28	45 days	SBM; ESB	60:40	nr	*7.8*	*−3.3*	*−7.5*	**−3.1**	[37]
		120 (in conc)	0.14 *	12 *					nr	*24.8*	*−5.6*	*−7.8*	*−1.1*	
Cow	PSCAKE	44	0.87 *	71 *	6	27 days	SBM	50:50	*−2.0*	*1.7*	*−0.9*	*0.3*	*0.2*	[40]
		83	1.64 *	133 *					*−2.0*	*3.5*	*−1.5*	*1.8*	*0.4*	
Cow	PSCAKE	56	1.37 *	194 *	6	21 days	SBM	50:50	−1.2	0.8	0.0	−1.6	0.6	[41]
		120	2.93 *	415 *					−2.2	1.1	0.9	−2.1	0.6	
Goat	PSCAKE	121 as fed (in conc)	0.14 *	15 *	36	28 days	SFM; C	nr	−0.5	*−3.1*	*5.3*	*7.2*	*−1.5*	[38]
Goat	PSCAKE	160 as fed (in conc)	0.16 as fed *	24 *	28	75 days	SBM; ESB	70:30	nr	*14.7*	*1.5*	*−3.1*	*0.7*	[39]
Goat	PSCAKE	160 as fed (in conc)	0.16 as fed *	nr	28	20 days	SBM; ESB	70:30	nr	*−18.7*	*−1.0*	*11.1*	*−1.6*	[65]
Cow	SFMEAL	70	1.50 *	20 *	12	21 days	SBM; WM	55:45	−2.7	−3.3	0.0	−2.7	2.3	[24]
		140	2.97 *	40 *					−3.6	−3.3	−3.1	−2.7	2.3	
		210	4.60 *	62 *					−0.5	−10.0	−3.1	−2.7	2.3	
Cow	SFMEAL	126 (in conc)	0.94 *	nr	12	21 days	SBM	nr	*3.3*	*−0.5*	*−1.0*	*−0.3*	*0.0*	[66]
Cow	SFMEAL	190 as fed (in conc)	nr	nr	18	67 days	MC; WB	60:40	*−* *0.9*	*−1.7*	*4.4*	*7.6*	*−14.3*	[72]
		380 as fed (in conc)	nr	nr					*0.2*	*−5.6*	*10.2*	*12.9*	*−14.3*	
Cow	SFMEAL	309 (in conc)	2.04 *	41 *	20	16 weeks	SBM; C	63:37	0.6	−0.5	0.0	−2.3	nr	[50]
Cow	SFMEAL	200	3.38 *	58 *	8	19 days	SBM; C	60:40	*−2.3*	*−5.9*	*−3.5*	*16.0*	*0.9*	[49]
Cow	SFMEAL	390 as fed (in conc)	4.64 as fed *	nr	24	21 days	SBM; SFH	49:51	−1.4	**−9.3**	*−1.9*	*−0.8*	*−0.2*	[68]
Cow	SFCAKE	150 (in conc)	nr	nr	16	15 days	SBM; C	nr	*1.6*	*−0.7*	*−0.6*	*−7.0*	*0.8*	[45]
Cow	SFCAKE	230 (in conc)	1.08	nr	10	63 days	SBM; PKM; DDG; HPF	75:25	*−1.0*	*−1.9*	*6.7*	*−7.5*	*2.1*	[67]
Cow	SFCAKE	80 *	2 as fed	255 *	32	3 weeks	SBM	nr	nr	*2.2*	**−2.8**	*−5.2*	nr	[46]
Sheep	SFMEAL	200	0.38 *	6 *	36	32 days	SBM; B	40:60	*0.5*	*−4.1*	nr	*11.7*	nr	[48]
		375	0.73 *	11 *					*3.2*	*−16.2*	nr	*2.2*	nr	
Sheep	SFCAKE	180 as fed (in conc)	0.11 as fed *	nr	30	75 days	SBM; C; B	nr	nr	nr	*−3.4*	**20.0**	*−0.2*	[69]
Sheep	SFCAKE	300 as fed (in conc)	0.20 *	nr	36	38 days	S	76:24	nr	*0.9*	*−7.0*	**−10.2**	nr	[44]
		500 as fed (in conc)	0.42 *	nr				66:34	nr	*−3.8*	*−3.1*	**−18.1**	nr	
Sheep	SFCAKE	560 (in conc)	0.51 *	nr	72	56 days	SBM; O; DDG; HPF	67:33	*1.5*	*4.9*	*4.6*	*−7.5*	*1.6*	[70]
		560 (in conc)	0.51 *	nr				50:50	*−1.6*	*9.1*	*2.1*	*−3.2*	*−0.2*	
Goat	SFMEAL	25 as fed *	0.2 as fed	nr	12	92 days	OM	87:13	nr	1.1	3.9	−1.6	0.2	[71]

Values in bold indicate significant differences (*p* < 0.05) compared to the control group; values in italic indicate non-significant differences (*p* > 0.05) compared to the control group; ^1^ HSCAKE—hempseed cake; PSCAKE—pumpkin seed cake; SFCAKE—sunflower cake; SFMEAL—sunflower meal; ^2^ Intake of fat from byproduct; ^3^ Main feed ingredient from control diet that was partially or totally replaced with byproduct (SFM—sunflower meal; C—corn; B—barley; SBM—soybean meal; ESB—extruded soybean; WM—wheat middlings; MC—mustard cake; WB—wheat bran; SFH—sunflower hulls; PKM—palm kernel meal; DDG—distilled dry grains; HPF—hydrogenated palm fat; S—soya; O—oats; OM—oat meal); ^4^ Dietary forage: concentrate ratio (on a dry matter basis); ^5^ Dry matter intake; ^6^ ((treatment value-control value)/control value) × 100; ^7^ in concentrate; *—calculated/estimated by the authors; nr—not reported.

**Table 3 animals-14-00539-t003:** The impact of incorporating hemp, pumpkin, and sunflower byproducts into the diets of dairy ruminants on milk fatty acid profile. The data are presented as the proportionate difference between the treatment group, considering the specific level of inclusion, and the control group.

Ruminant Species	Byproduct Form ^1^	Inclusion in Diet (g/kg DM Unless Otherwise Stated)	Intake of BP (kg DM/Day Unless Otherwise Stated)	Intake of Fat (g/Day) ^2^	SFA ^3^	MUFA ^4^	PUFA ^5^	n-6 ^6^	n-3 ^7^	n-6/n-3	RA ^8^	VA ^9^	Reference
					% Change—Relative to Control ^10^								
Sheep	HSCAKE	213 as fed *	0.48 as fed	50 *	**−12.3**	**14.4**	**51.9**	*14.2*	**47.7**	**−22.5**	**146.7**	**100.8**	[36]
Goat	HSCAKE	60 (in conc) ^11^	0.07 *	6 *	*5.4*	**−9.6**	*4.8*	*1.1*	*11.7*	−9.5 *	nr	nr	[37]
		120 (in conc)	0.14 *	12 *	*4.5*	*−7.0*	*−2.3*	*−4.8*	*13.3*	−16.1 *	nr	nr	
Cow	PSCAKE	56	1.37 *	194 *	2.1 *	0.3 *	−2.2 *	−2.7 *	0.0 *	−2.6 *	−7.0	nr	[41]
		120	2.93 *	415 *	0.7 *	2.1 *	3.8 *	4.6 *	0.0 *	4.7 *	16.3	nr	
Goat	PSCAKE	121 as fed (in conc)	0.14 *	15 *	**−5.0**	**12.1**	**24.4**	**30.0**	*3.0*	**28.0**	*10.5*	nr	[38]
Goat	PSCAKE	160 as fed (in conc)	0.16 as fed *	24 *	*1.7*	*−1.7*	*−15.3*	**−16.4**	*−9.7*	−7.4 *	*−15.6*	nr	[39]
Goat	PSCAKE	160 as fed (in conc)	0.16 as fed *	nr	*1.5*	*0.5*	**−17.9**	**−17.9**	**−24.6**	9.0 *	*−16.8*	*−22.6*	[65]
Cow	SFCAKE	230 (in conc)	1.08	nr	*−1.3*	*1.8*	**14.8**	9.1 *	3.5 *	*5.2*	**35.4**	**31.9**	[67]
Cow	SFCAKE	80 *	2 as fed	255 *	**−3.0**	**6.2**	**11.6**	nr	*−2.6*	nr	**34.0**	**39.1**	[46]
Sheep	SFCAKE	180 as fed (in conc)	0.11 as fed *	nr	**−4.6**	**10.3**	*15.5*	**23.4**	*−8.0*	**43.7**	nr	nr	[69]
Sheep	SFCAKE	300 as fed (in conc)	0.20 *	nr	**−5.4**	*5.3*		nr	nr	nr	**11.2**	**9.7**	[44]
		500 as fed (in conc)	0.42 *	nr	**−22.2**	**43.6**		nr	nr	nr	**150.2**	**163.1**	
Sheep	SFCAKE	560 (in conc)	0.51 *	nr	**−12.2**	*9.6*	**134.5**	**204.6**	**36.6**	**124.0**	**78.8**	**71.0**	[70]
		560 (in conc)	0.51 *	nr	**−14.6**	**14.9**	**155.8**	**204.4**	**39.1**	**110.2**	**155.6**	**182.4**	

Values in bold indicate significant differences (*p* < 0.05) compared to the control group; values in italic indicate non-significant differences (*p* > 0.05) compared to the control group; ^1^ HSCAKE—hempseed cake; PSCAKE—pumpkin seed cake; SFCAKE—sunflower cake; ^2^ Intake of fat from byproduct; ^3^ SFA—saturated fatty acids; ^4^ MUFA—monounsaturated fatty acids; ^5^ PUFA—polyunsaturated fatty acids; ^6^ n-6—omega 6 fatty acids; ^7^ n-3—omega 3 fatty acids; ^8^ RA—rumenic acid; ^9^ VA—vaccenic acid; ^10^ ((treatment value-control value)/control value) × 100; ^11^ in concentrate; *—calculated/estimated by the authors; nr—not reported.

## 6. Effect of Camelina and Linseed Byproducts on Milk Yield, Composition, and Fatty Acids

### 6.1. Milk Yield and Composition

The results presented in Table 4 clearly demonstrate that the inclusion of camelina seed BPs in dairy ruminant diets leads to a reduction in milk fat content, regardless of the form of the BP. When feeding diets with high levels of PUFA, the ruminal biohydrogenation of PUFA leads to the formation of intermediates such as C18:1 trans-10 and trans-10, cis-12 CLA, representing a shift from the trans-11 to the trans-10 pathway of FA biohydrogenation [28,55,56]. These intermediates are potent inhibitors of *de novo* milk fat synthesis when they reach the mammary gland [5]. Mutsvangwa, et al. [56] suggested that the observed reduction in milk fat content was probably attributable to these intermediates. Additionally, Sarramone, et al. [28] emphasized that the process of obtaining BPs, such as press extraction, which is known to break down the cell walls of the seed, makes its residual oil more readily available. The authors put forth the hypothesis that the increased oil availability in camelina seed expeller (CSEXP) might lead to ruminal disturbances, potentially prompting a transition from the trans-11 to the trans-10 pathway in FA biohydrogenation. Based on that, Sarramone, et al. [28] proposed that a more thorough delipidation process, achieved through solvent extraction rather than pressing technologies, might be a more effective approach to mitigate the adverse effects of readily available free oil on lactation efficiency. In addition to the decrease in milk fat content, none of the studies reported significant alterations in milk yield, as well as in protein and lactose content.

Research investigating the feasibility of incorporating linseed BPs into ruminant diets has produced less consistent findings compared to camelina BPs (Table 4). In the majority of research investigations, the addition of linseed BPs to the diets of both cows [58,59,64,74,75,76,77] and goats [61] did not result in significant changes in milk yield, protein, or fat content. Conversely, Jozwik, et al. [57] and Sayed, et al. [60] documented a significant increase in milk yield, attributing it to the higher energy content of the diet incorporating LSC. Brito, et al. [47] and Khorasani, et al. [62] both noted a substantial reduction in cow milk fat content, likely resulting from the inhibition of mammary lipid synthesis caused by specific FA intermediates generated during the ruminal biohydrogenation of dietary PUFAs. Alongside the decline in milk fat content, Brito, et al. [47] also reported a decrease in milk yield, which was explained with less energy dense diet containing linseed meal (LSMEAL) and an increase in milk protein content, which can be attributed to the reduction in milk volume.

**Table 4 animals-14-00539-t004:** The impact of incorporating camelina and linseed byproducts into the diets of dairy ruminants on dry matter intake, milk yield, protein percentage, fat percentage, and lactose percentage. The data are presented as the proportionate difference between the treatment group, considering the specific level of inclusion, and the control group.

Ruminant Species	Byproduct Form ^1^	Inclusion in Diet (g/kg DM Unless Otherwise Stated)	Intake of BP (kg DM/Day Unless Otherwise Stated)	Intake of Fat (g/Day) ^2^	Number of Animals	Trial Duration	Main Feed Ingredient Replaced ^3^	F:C ^4^	DMI ^5^	Milk Yield	Milk Protein	Milk Fat	Milk Lactose	Reference
									% Change—Relative to Control ^6^					
Cow	CSCAKE *	50	0.91 *	117 *	24	nr	SFM	70:30	0.1	*−4.8*	*2.1*	*−6.9*	*−0.1*	[54]
		100	1.83 *	236 *					0.2	*1.9*	*3.5*	*−6.9*	*−0.1*	
Cow	CSCAKE *	95	1.88 *	248 *	6	4 weeks	SBM; energy concentrate	58:42	*−5.7*	*−3.8*	*−4.5*	**−54.1**	*−3.7*	[53]
Cow	CSEXP	95	2.32 *	nr	4	21 days	DDG	48:52	*3.0*	*−1.1*	*−2.8*	**−25.3**	*0.6*	[28]
Cow	CSEXP	200 as fed (in conc) ^7^	2.38 as fed *	350	5	21 days	RSM	55:45	*−2.6*	*3.5*	*−4.5*	*−7.3*	*0.2*	[78]
Cow	CSEXP	50	1.36 *	201 *	8	28 days	CM	45:55	0.7	10.5	−3.5	−7.4	0.2	[56]
		75	2.02 *	299 *					0.0	5.1	−6.1	−14.4	−0.5	
		100	2.67 *	395 *					−0.7	9.4	−5.7	−14.1	0.2	
Cow	CSCAKE	nr	nr	500	4	18 days	SBM	nr	nr	−11.8	−3.2	−17.2	−0.6	[77]
Sheep	CSCAKE	30	0.1 *	14 *	30	110 days	RSM	70:30	nr	*11.4*	*−1.7*	*−5.7*	*229.9*	[55]
		60	0.2 *	28 *					nr	*7.7*	*−0.8*	**−14.2**	*222.1*	
Goat	CSCAKE	120 (in conc)	0.12 *	17 *	66	nr	RSM	70:30	nr	nr	−0.3	−0.6	−1.1	[52]
Cow	LSCAKE	3 as fed *	0.3 as fed	nr	40	7 weeks	-	99:1	nr	nr	*−0.7*	*0.5*	*0.2*	[74]
Cow	LSMEAL	160 (in conc)	0.79 *	11 *	16	19 days	SBM; SFM	70:30	*0.6*	**−6.8**	**5.4**	**−3.0**	**−0.4**	[47]
Cow	LSMEAL	124	4.04 *	52 *	8	21 days	SBM; BP	60:40	**3.5**	*5.4*	*1.1*	*−4.0*	*0.0*	[64]
Cow	LSCAKE *	90 (in conc)	0.61 *	52 *	16	180 days	SBM; SFM; WB	50:50	*4.0*	**27.7**	*−1.1*	*0.3*	*0.2*	[60]
Cow	LSMEAL	246 (in conc)	1.73 *	123 *	4	21 days	CM	60:40	*−0.6*	*13.4*	*−2.3*	**−15.1**	*3.5*	[62]
Cow	LSEXP	48	0.97 *	nr	8	21 days	SBM; C; BP	60:40	−3.3	2.8	−1.7	−3.3	2.4	[75]
		95	2.01 *	nr					0.5	1.4	−0.8	−7.9	1.8	
		141	3.10 *	nr					4.8	1.4	−1.4	−1.6	1.6	
Cow	LSMEAL	94	1.90 *	nr	24	4 weeks	B; C; PS	47:53	*−5.6*	*−5.3*	*8.4*	*8.9*	*−0.4*	[76]
Cow	LSCAKE	32 *	0.46	73	8	21 days	RSC	65:35	1.9	−1.6	0.9	2.8	−1.2	[59]
		64 *	0.89	143					−3.2	−7.0	−1.2	6.4	−0.2	
		92 *	1.29	207					−1.3	−7.6	−3.0	7.8	−0.2	
Cow	LSCAKE	213 (in conc)	1.57 *	210 *	16	70 days	RSC	57:43	*5.5*	*5.5*	*5.8*	*8.7*	**−3.4**	[58]
Cow	LSCAKE	nr	nr	500	4	18 days	SBM	nr	nr	12.7	−7.7	−6.3	−0.8	[77]
Goat	extruded LSCAKE	50	0.1	16	30	21 days	P	nr	nr	*−12.5*	*−0.7*	*3.9*	nr	[61]
		100	0.2	32					nr	*−4.2*	*4.9*	*5.3*	nr	
Goat	LSCAKE	199	nr	nr	16	5 weeks	RSM; T	60:40	nr	**27.3**	nr	nr	nr	[57]

Values in bold indicate significant differences (*p* < 0.05) compared to the control group; values in italic indicate non-significant differences (*p* > 0.05) compared to the control group; ^1^ CSCAKE—camelina seed cake; CSEXP—camelina seed expeller; LSCAKE—linseed cake; LSMEAL—linseed meal; LSEXP—linseed expeller; ^2^ Intake of fat from byproduct; ^3^ Main feed ingredient from control diet that was partially or totally replaced with byproduct (SFM—sunflower meal; SBM—soybean meal; DDG—distilled dry grains; RSM—rapeseed meal; CM—canola meal; BP—beet pulp; WB—wheat bran; C—corn; B—barley; PS—protein supplement; RSC—rapeseed cake; P—peas; T—triticale); ^4^ Dietary forage: concentrate ratio (on a dry matter basis); ^5^ Dry matter intake; ^6^ ((treatment value-control value)/control value) × 100; ^7^ in concentrate; *—calculated/estimated by the authors; nr—not reported.

Clearly, the differences in chemical composition between the control and experimental diets significantly affected milk production. As previously mentioned, this impact on milk production traits might be a result of variations in composition rather than solely the effect of the oilseed BPs.

### 6.2. Milk Fatty Acids

The inclusion of camelina seed BPs in dairy ruminant nutrition led to desirable changes in milk FA composition in terms of SFA, MUFA, and PUFA proportion regardless of the BP form (Table 5). A decrease in milk SFAs and an increase in MUFAs and PUFAs were reported in cows [28,53,54,56,77,78], sheep [55], and goats [52]. Simultaneously, there was an observed increase in milk n-3 FA, consequently leading to the desirable lower n-6/n-3 ratio. Improvement in milk RA and VA content was reported in all studies. The increase in both milk RA and VA, when compared to the control diet without camelina BPs, ranged from approximately 70% to over 600%. The best results were documented in the study by Mihhejev, et al. [77], possibly due to the high inclusion level of camelina BPs and concurrent high camelina oil in the dairy cows’ diet, but this result has not been confirmed in any subsequent study.

The inclusion of linseed BPs in dairy cow [58,64,74,76,77] and goat [57,61] nutrition led to the improvement in milk FA profile, which is same as in the case of camelina seed BPs (Table 5). The addition of LSMEAL and linseed cake (LSCAKE) lowered SFAs and increased MUFAs and PUFAs in ruminant milk. Concurrently, milk n-3 FA increased, consequently leading to the desirable lower n-6/n-3 ratio. The improvement in milk RA ranged from 40 up to 95%, while for VA it ranged from 20 up to 100%. Only one study reported a decrease in RA and VA. In the Brito, et al. [47] study, soybean meal and SFMEAL were completely replaced with LSMEAL. This substitution, attributed to low fat content in LSMEAL, resulted in a decrease in fat content in the experimental concentrate mix to 1.88%, as opposed to the 5.19% in the control concentrate mix. The authors suggested that milk FA profile was changed likely as a result of differences in FA intake, Δ9-desaturase indices, and ruminal biohydrogenation pathways.

A consistent trend in improvement of the milk FA profile was observed when animals were fed camelina, linseed, and sunflower BP-supplemented diets. Nonetheless, many of the observed responses in studies incorporating oilseed industry BPs might be associated with variations in chemical composition of experimental and control diet (presented in Appendix A). Altman, et al. [79] suggested that the directly proportional changes in milk FA composition may be more linked to differences in fat supplementation rates in the treatment diets rather than representing a novel effect of BP supplementation. Nevertheless, the gathered results suggest the potential value of oilseed industry BPs, further leading to the conclusion that the use of BPs in ruminant nutrition should not be neglected. Additional further research is necessary before reaching definite conclusions.

**Table 5 animals-14-00539-t005:** The impact of incorporating camelina and linseed byproducts into the diets of dairy ruminants on milk fatty acid profile. The data are presented as the proportionate difference between the treatment group, considering the specific level of inclusion, and the control group.

Ruminant Species	Byproduct Form ^1^	Inclusion in Diet (g/kg DM Unless Otherwise Stated)	Intake of BP (kg DM/Day Unless Otherwise Stated)	Intake of Fat (g/Day) ^2^	SFA ^3^	MUFA ^4^	PUFA ^5^	n-6 ^6^	n-3 ^7^	n-6/n-3	RA ^8^	VA ^9^	Reference
					% Change—Relative to Control ^10^								
Cow	CSCAKE *	50	0.91 *	117 *	**−4.8**	**6.9**	**16.5**	**18.3**	*3.3*	14.5 *	nr	nr	[54]
		100	1.83 *	236 *	**−7.1**	**10.2**	**32.4**	**35.0**	*11.7*	20.9 *	nr	nr	
Cow	CSCAKE *	95	1.88 *	248 *	**−21.0**	**58.2**	**35.2**	nr	nr	nr	**81.4**	**165.1**	[53]
Cow	CSEXP	95	2.32 *	nr	−11.5 *	28.2 *	22.4 *	−34.4 *	27.0 *	−48.4 *	**101.2**	**61.4**	[28]
Cow	CSEXP	200 as fed (in conc) ^11^	2.38 as fed *	350	−11.8	28.0	36.4	nr	nr	nr	131.3	127.1	[78]
Cow	CSEXP	50	1.36 *	201 *	−7.2	18.6	7.8	0.3	14.3	−12.0	88.0	nr	[56]
		75	2.02 *	299 *	−14.5	35.3	27.8	15.4	40.8	−19.9	124.0	nr	
		100	2.67 *	395 *	−17.7	43.4	34.5	19.8	44.9	−14.9	176.0	nr	
Cow	CSCAKE	nr	nr	500	−31.3	35.6	319.5	nr	nr	nr	675.0	640.2	[77]
Sheep	CSCAKE	30	0.1 *	14 *	**−7.0**	**11.9**	**23.3**	*11.7*	**45.8**	**−21.9**	**72.7**	**68.9**	[55]
		60	0.2 *	28 *	**−15.3**	**36.6**	**34.8**	*11.0*	**78.5**	**−36.6**	**78.3**	**144.6**	
Goat	CSCAKE	120 (in conc)	0.12 *	17 *	**−11.6**	**16.0**	**64.4**	*14.3*	**71.0**	−33.1 *	nr	**579.3**	[52]
Cow	LSCAKE	3 as fed *	0.3 as fed	nr	−8.1 *	14.4 *	39.8 *	42.2 *	29.8 *	9.5 *	**94.3**	**59.2**	[74]
Cow	LSMEAL	160 (in conc)	0.79 *	11 *	**6.8**	**−15.8**	**−17.2**	**−33.8**	*3.7*	**−38.7**	**−18.4**	**−25.9**	[47]
Cow	LSMEAL	124	4.04 *	52 *	*−0.2*	*1.2*	*−4.1*	**−9.9**	*14.7*	**−22.7**	nr	nr	[64]
Cow	LSMEAL	94	1.90 *	nr	nr	*13.7*	*19.9*	*4.5*	**69.0**	**−38.4**	*38.6*	*20.0*	[76]
Cow	LSCAKE	213 (in conc)	1.57 *	210 *	*−5.2*	**12.7**	*15.7*	**6.0**	**97.0**	**−46.2**	nr	nr	[58]
Cow	LSCAKE	nr	nr	500	−10.8	19.0	65.9	nr	nr	nr	45.3	58.9	[77]
Goat	extruded LSCAKE	50	0.1	16	−5.9 *	13.5 *	20.5 *	13.0 *	67.3 *	−32.5 *	**52.4**	**75.7**	[61]
		100	0.2	32	−10.4 *	26.3 *	27.2 *	15.9 *	101.8 *	−42.5 *	**66.7**	**98.6**	
Goat	LSCAKE	199	nr	nr	**−20.3**	**29.0**	*−27.9*	nr	nr	nr	nr	nr	[57]

Values in bold indicate significant differences (*p* < 0.05) compared to the control group; values in italic indicate non-significant differences (*p* > 0.05) compared to the control group; ^1^ CSCAKE—camelina seed cake; CSEXP—camelina seed expeller; LSCAKE—linseed cake; LSMEAL—linseed meal; LSEXP—linseed expeller; ^2^ Intake of fat from byproduct; ^3^ SFA—saturated fatty acids; ^4^ MUFA—monounsaturated fatty acids; ^5^ PUFA—polyunsaturated fatty acids; ^6^ n-6—omega 6 fatty acids; ^7^ n-3—omega 3 fatty acids; ^8^ RA—rumenic acid; ^9^ VA—vaccenic acid; ^10^ ((treatment value-control value)/control value) × 100; ^11^ in concentrate; *—calculated/estimated by the authors; nr—not reported.

## 7. Benefits, Drawbacks, and Future Research Aspects

Dietary utilization of oilseed BPs in ruminant diets can have extensive benefits on several levels. The use of BPs in the winter season enables the provision of essential nutrients to animals comparable to those supplied by pasture grass in the summer. This practice results in the enrichment of milk with valuable bioactive compounds, thus enhancing the overall nutritional content [74]. From an economic perspective, feeding ruminants with these BPs offers cost-effective feed alternatives, as they are available at a relatively lower cost compared to some conventional feed ingredients, thereby reducing production costs [49]. In addition, oilseed BPs (particularly cake and expeller) can be generated on the farm, presenting a viable local alternative that not only contributes to the advocacy of low-input production systems but also plays a role in diminishing greenhouse gas emissions from feed transport [67]. Furthermore, an increased utilization of the byproducts creates a more resilient and sustainable ruminant sector by reducing food–feed competition and the problem of waste, promoting resource efficiency, and mitigating environmental impact [80].

Nevertheless, the application of oilseed BPs in ruminant feeding is hindered by an inconsistency in chemical composition, which can vary markedly depending on the production process. The feed production industry prefers feed ingredients of consistent composition and quality, as well as unceasing supply.

Additional investigation should be undertaken to enhance the current understanding in this domain, with a specific emphasis on (1) the effect of long-term feeding of selected oilseed BPs on the performance of dairy ruminants; (2) the impact of dietary oilseed BPs supplementation on animal health and welfare; (3) the effect of alterations in milk FA profiles on the shelf-life, quality, and sensory attributes of dairy products such as cheese, butter, yogurt, and kefir; (4) establishing the optimal inclusion levels of each oilseed BP in the rations of distinct dairy ruminant species. Furthermore, these studies should be rigorous in methodologies with a specific focus on mitigating potential confounding influences arising from variations in diet chemical composition. This ensures drawing accurate conclusions that reflect the true potential of oilseed BPs application in ruminant diet.

## 8. Conclusions

The present review demonstrates that the inclusion of selected oilseed industry BPs rich in LA (hemp, pumpkin seed, and sunflower) and ALA (camelina and linseed) in the nutrition of dairy ruminants generally does not lead to any detrimental effects on milk production and composition. A consistent pattern of enhancement in the milk FA profile was noted when animals were provided diets supplemented with camelina, linseed, and sunflower BPs. On the other hand, no changes were pronounced in observed parameters when pumpkin BPs were included in ruminant diets. Although the presented studies have indicated that hempseed BPs have potential value as an alternative ingredient in dairy ruminant diet, more studies on their utilization are needed.

It is noteworthy to mention the certain discrepancies in the chemical composition of the diets in the abovementioned collected papers, making it challenging to draw valid conclusions regarding the impact of the dietary utilization of observed BPs in dairy ruminants. A wider range of research within each ruminant species is required to establish and validate the true value and impact of their use in the ruminant diet. To confirm that the altered milk production traits and FA profile are attributed to the characteristic composition of FA in the presented oilseed BPs, these investigations should be structured in a way to mitigate the potential perplexing impact of variations in dietary formulation. This entails, among other things, ensuring a consistent chemical composition of the diets, particularly in terms of dietary fat content.

## Figures and Tables

**Figure 1 animals-14-00539-f001:**
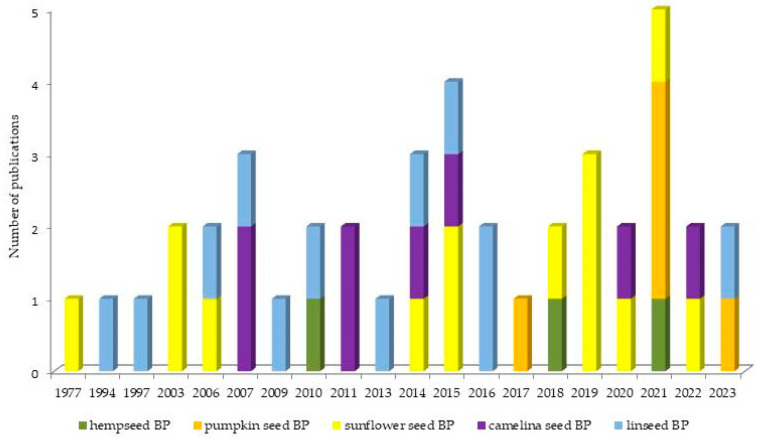
Overview of the chronological progression of publications over the years.

**Figure 2 animals-14-00539-f002:**
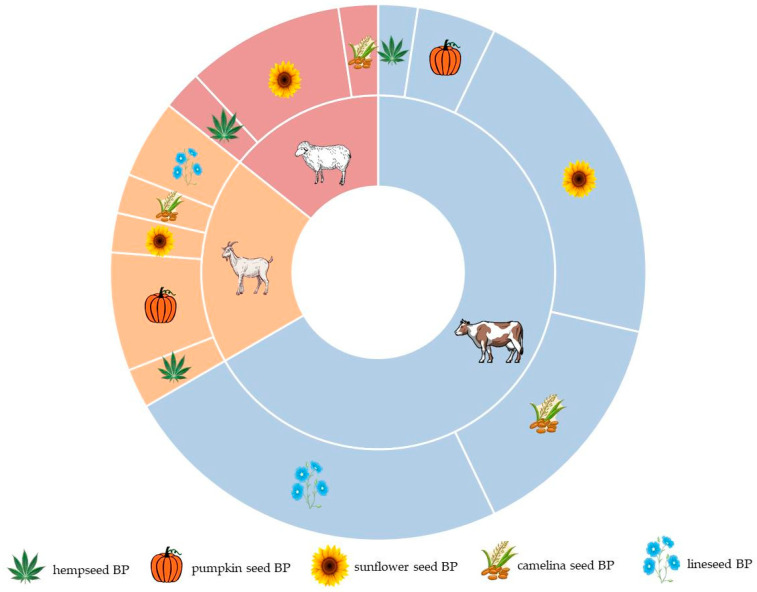
Distribution of ruminant species and byproducts from the oilseed industry among the selected studies.

## Data Availability

No new data were created or analyzed in this study. Data sharing is not applicable to this article.

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
