# Peer review of "Impact of Using Oilseed Industry Byproducts Rich in Linoleic and Alpha-Linolenic Acid in Ruminant Nutrition on Milk Production and Milk Fatty Acid Profile"

_animals, 2024, doi:10.3390/ani14040539_

Round 1
Reviewer 1 Report
Comments and Suggestions for Authors
The authors are to be commended for collecting information to review these byproducts in ruminant diets. However, I feel their review falls short for several reasons.
First, the oil seed byproducts selected are minor components in EU feeds, according to information published by the EU. The EU reports that the most oil seed byproduct consumed in animal feeds is from soybeans. Of the meals used in animal feeds 55% is from soybeans, 23% from rapeseed, 15% from sunflower, and 7% from other meals. Thus the paper is reporting on meals with little inclusion across EU animal feeds, except for sunflower meal. Therefore the paper's conclusion would have little interest both due to the few studies for hemp and pumpkin and the lack of significant inclusion in EU animal feeds. If the intent was to make a case to use these oil seed byproducts in lieu of soybean or rapeseed, then including a comparison to these meals would be important. In fact in many of the studies sited the effect reported was the substitution of the oil seed byproduct in comparison to soybean meal. Including soybean meal in the report would contribute to completeness of the study.
Secondly, comparing across sheep, goat, and cows is a problem especially with the few studies within species and meals.
Areas of weakness: The distribution of animals and oilseed products is as follows:
Sunflower Linseed Hemp Camelina Pumpkin
Goat 1 2 1 1 3
Sheep 3 0 1 1 0
Cows 10 9 1 6 2
Total 14 11 3 8 5
Concern with comparing species with differences in milk fat content, milk production, feed intake, rumen turnover, and lactation stages and only a few observations for some byproducts within species.
The content of byproduct inclusion in the various diets and the basal diet are quite different. Results across species would be extremely difficult to generalize.
The tables are extremely difficult to follow for formatting and also for how the data is reported.
Table 1-4. The tables are very difficult to read due to the formatting and column breaks and wrap around text. Possibly putting tables on a landscape orientation would help solve the column issues.
The results in the tables are very confusing as they are reported relative to the control group. It would be more informative to this reader to report the absolute numbers with coefficients of significance and standard errors for each group and diet. The changes are not clear - are they differences or are they proportional differences? Report the actual data by group and diet. This would provide the actual production information for each group.
How the data is presented makes it very difficult to follow the trends the authors discuss in the text. Re-doing the tables in a clearer presentation would be of great benefit.
specific line comments follow
Lines 19-22 versus lines 34-36 These statements seem contradictory.
19-22 Simple summary: Concluding sentence: "This review concludes that, overall, the selected oilseed byproducts do not have adverse effects on milk production and composition, with camelina and linseed byproducts demonstrating the most promising potential to improve the fatty acid profile of milk."
34-36 Abstract: Concluding sentence: "Sunflower and hempseed byproducts demonstrate considerable potential in influencing the profile of milk fatty acids."
Line 269 "pursued up today" should be pursued up to today
Line 290 the protein content reported is confusing - it seems to be for the total diet, but the way the sentence is worded one expects protein supply from the hemp cake.
Lines 286-306 Discussion of hemp. The inclusion rate of hemp should be reported for the three studies sited. Otherwise the reader cannot make sense of the varying results.
Lines 308-314 Pumpkin see cake studies. Here a comparison to soybean and sunflower is referred to. More specific statements on milk production, fat and fatty acid content of milk would be more informative than the general statement "no detrimental effects were observed on milk production or quality." Quality could mean many things. More specific information is needed.
Lines 316-328. Sunflower BP. Schingoethe's paper would be better to be worded that replacing soybean meal with sunflower meal had no significant effect on milk yield and milk fat and protein content. This is the important observation in their study; not soybean meal replacing sunflower meal.
Line 344 "SFA, which are primarily derived from ruminal microflora" This statement is too restrictive, as the mammary gland produces in novo about 50% of fatty acids in milk, which are primarily SFA of varying chain length.
Reviewer 2 Report
Comments and Suggestions for Authors
I would like to thank the Editorial Staff of Animals for entrusting me with the review of the Article Manuscript ID: Animals- 2756188 entitled: Title: Improvement of Milk Fatty Acid Profile Using Oilseed Industry Byproducts Rich in Linoleic and Alpha-linolenic Acid
Authors: Bojana Kokić, Slađana Rakita, Jelena Vujetić
Submitted to section: Animal Nutrition,
Dear authors,
The manuscript is interesting, well written and discussed
This review provides a comprehensive overview of the existing research studies regarding the incorporation of hemp, pumpkin, sunflower, camelina and linseed byproducts into the nutrition of dairy cows, sheep and goats, their influence on milk production traits and milk fatty acid composition.
P:80-83 It should be developed more the ruminants
P:93-94 Unintelligible; numerous studies and only one item of literature is given
P:114-115 A loose statement not supported by research
P:117-123 Why only those products of the oil industry were compared?
Chapter 2 is loosely related to the title of the work. The focus should be on the possibility of producing and using oilseed crops and waste from the oil industry in animal (ruminant) nutrition.
Table 1 is not very clear
P:278-279 It seems that 41 items of literature were suggested while in the years analyzed there are many more studies involving oil crops
Table 2, 3, 4 and 5 Is not very clear
The summary is very general. While it is based on a review of research in the field, it does not give a clear answer for practical use as to which components are better. Attention should be paid to the direction of health values in ruminant milk in terms of the consumer. Analogous to the purpose of the work and the Introduction.
In conclusion, I state that due to the current topic regarding the search for dietary modifying ingredients and the health benefits of milk, the paper should be published after taking into account the comments.
Best regards
Reviewer 3 Report
Comments and Suggestions for Authors
In the review "Improvement of Milk Fatty Acid Profile Using Oilseed Industry Byproducts Rich in Linoleic and Alpha-linolenic Acid" the authors Kokić, Rakita and Vujetić summarize the literature from 41 research studies on the utilization of oilseed industry byproducts rich in linoleic acid and alpha-linolenic acid in dairy cows, sheep and goats nutrition and their impact on milk production characteristics, and potential to improve fatty acid composition of milk. The review is well summarized and the tables and figures are easy to understand the literature done so far. However, this review looks more like a summary of the literature. It would be interesting to the readers if important questions concerning the field is asked and promising directions of research needed is mentioned in a separate paragraph.
Comments on the Quality of English LanguageMinor editing of English language required
Round 2
Reviewer 1 Report
Comments and Suggestions for Authors
Line 62-63: "The next significant transformation of dietary fats in the rumen is the biohydrogenation of unsaturated fatty acids (UFA) into saturated fatty acids (SFA), particularly stearic acid."
I suggest modifying this sentence, as stearic acid does predominate, but only if C-18 unsaturated fatty acids are the main fatty acid in the diet. The major SFA in bovine milk is palmitic acid (Jensen JDS 1991).
I suggest changing the sentence to indicate stearic acid is formed from PUFA C18 fatty acids, which are predominate in plant oils.
Line 306 and Table 1. Source(s) of dietary compositions?
Line 474 should this read "sunflower meal substituting for soybean meal'? It seems the study was to investigate sunflower meal replacing soybean meal in 10 lactating cows rations.
Line 482 "Additionally, in Oliveira, et al. [24] study, an increase in the proportion of SFMEAL led to a linear decrease in milk yield." Any differences in the amount fed, production stage of the cows, or other characteristics of the diet that would contribute to Oliveira finding a reduction in milk yield compared to Schingoethe's observations and the other papers sited?
Table 2-4. suggest you place a formula in the foot note as to the calculation of the proportion in general form such as ((treatment value - control value)/control value) x 100. I still find the tables extremely confusing and difficult to follow. For example DMI, your footnote says dry matter intake. The reader expects it to be kg/head/day, typical for production papers. Instead this may be a proportionate change.
It would also help to provide a heading over the items such as
% change -relative to control--------------------------------------------------------
DMI Milk Yield Milk protein Milk fat Milk lactose
Line 636 "Dietary utilization of oilseed BP in ruminant diet can have extensive benefits on 636 several levels. The use of BP in winter season enables the provision of essential nutrients 637 to animals comparable to those supplied by pasture grass in the summer. This practice 638 results in the enrichment of milk with valuable bioactive compounds, thus enhancing 639 overall nutritional content. From an economic perspective, feeding ruminants with these 640 BP offers cost-effective feed alternatives, as they are available at a relatively lower cost 641 compared to some conventional feed ingredients, thereby reducing production costs. In 642 addition, oilseed BP (particularly cake and expeller) can be generated on the farm, pre-643 senting a viable local alternative that not only contributes to the advocacy of low-input 644 production systems but also plays a role in diminishing greenhouse gas emissions from 645 feed transport."
These conclusions are not in the paper, nor were they investigated. They may be true, but they are opinion and would need a different analysis then presented here.
Overall: Basically the paper demonstrates these byproducts can be fed to ruminants. There is an imbalance in species and byproduct fed, so conclusions on effects on production and fatty acid content are very preliminary, as a more complete description of diets would be needed. Such as fat content, which is presented in some, but also non-fiber carbohydrate, as starch content has been shown to influence the effects on bio-hydrogenation and milk fat depression.
I think the paper still needs significant improvement. The composition of feeds in table 1 - were they from the papers referenced or from other sources? If not from the papers referenced then they may not apply to the studies sited.
I still hesitate to accept the paper. Why, you probably ask. You should better characterize the diets used in the studies. Especially the non-fiber carbohydrate content would be important as to the influence on the PUFA in the diet and milk fat. For example, just a rough examination of the studies on milk fat, 44 of the groups fed had a lower fat % compared to 24% of the observations with a positive increase in fat%. Why? Protein % was lower across the majority of groups fed. it could be a typical effect of fat in diets, or it could be the rumen undegradable protein was not sufficient with the fat inclusion. Milk production and dry matter intake increased in roughly half the groups and decreased in roughly half the groups, therefore these probably were not influenced by the BP inclusions. Although only 24 of the 41 studies had fatty acid profiles in milk, there was a consistent trend to increase MUFA, PUFA, and RA and VA.
It would be more appropriate to present the data based on specie rather than by-product. A table for cows and by-products; a table for goats and by-products; etc. The reason is the difference in feed intake and dietary ingredients probably are quite different across the species.
Reviewer 3 Report
Comments and Suggestions for Authors
The authors have addressed my concerns satisfactorily.
Author Response
The authors express sincere gratitude for the useful suggestions and insightful comments provided.